# Inherited Reticulate Pigmentary Disorders

**DOI:** 10.3390/genes14061300

**Published:** 2023-06-20

**Authors:** Min-Huei Lin, Pei-Chen Chou, I-Chen Lee, Syuan-Fei Yang, Hsin-Su Yu, Sebastian Yu

**Affiliations:** 1School of Medicine, College of Medicine, Kaohsiung Medical University, Kaohsiung 807, Taiwan; u110001147@gap.kmu.edu.tw (M.-H.L.); u110001005@gap.kmu.edu.tw (P.-C.C.); u110001132@gap.kmu.edu.tw (I.-C.L.); ggemstone02@gmail.com (S.-F.Y.); 2Department of Dermatology, College of Medicine, Kaohsiung Medical University, Kaohsiung 807, Taiwan; yup.kmu@gmail.com; 3Graduate Institute of Clinical Medicine, College of Medicine, Kaohsiung Medical University, Kaohsiung 807, Taiwan; 4Department of Dermatology, Kaohsiung Medical University Hospital, Kaohsiung 807, Taiwan; 5Neuroscience Research Center, Kaohsiung Medical University, Kaohsiung 807, Taiwan

**Keywords:** dermatopathia pigmentosa reticularis, Dowling-Degos disease, dyschromatosis symmetrica hereditaria, dyschromatosis universalis hereditaria, dyskeratosis congenita, Naegeli–Franceschetti–Jadassohn syndrome, reticulate acropigmentation of Kitamura, X-linked reticulate pigmentary disorder

## Abstract

Reticulate pigmentary disorders (RPDs) are a group of inherited and acquired skin conditions characterized by hyperpigmented and/or hypopigmented macules. Inherited RPDs include dyschromatosis symmetrica hereditaria (DSH), dyschromatosis universalis hereditaria (DUH), reticulate acropigmentation of Kitamura (RAK), Dowling-Degos disease (DDD), dyskeratosis congenita (DKC), Naegeli–Franceschetti–Jadassohn syndrome (NFJS), dermatopathia pigmentosa reticularis (DPR), and X-linked reticulate pigmentary disorder. Although reticulate pattern of pigmentation is a common characteristic of this spectrum of disorders, the distribution of pigmentation varies among these disorders, and there may be clinical manifestations beyond pigmentation. DSH, DUH, and RAK are mostly reported in East Asian ethnicities. DDD is more common in Caucasians, although it is also reported in Asian countries. Other RPDs show no racial predilection. This article reviews the clinical, histological, and genetic variations of inherited RPDs.

## 1. Introduction

Reticulate pigmentary disorders are a heterogeneous group of skin conditions characterized by hyperpigmented and/or hypopigmented macules with varying extent of pigment and size [1]. The term reticulate describes freckle-like pigmentary lesions with indistinct borders [2]. Reticulate pigmentary disorders encompass skin conditions that are inherited and acquired. Inherited reticulate pigmentary disorders are caused by various genetic mutations and have distinct clinical manifestations, such as age of onset and distribution of the pigmentation. This article reviews clinical, histological, and genetic variations of inherited reticulate pigmentary disorders, including dyschromatosis symmetrica hereditarian (DSH), dyschromatosis universalis hereditarian (DUH), reticulate acropigmentation of Kitamura (RAK), Dowling-Degos disease (DDD), dyskeratosis congenita (DKC), Naegeli–Franceschetti–Jadassohn syndrome (NFJS), dermatopathia pigmentosa reticularis (DPR), and X-linked reticulate pigmentary disorder (XLRPD).

## 2. Search Strategy

In this narrative review, we conducted a search for publications in the PubMed, Embase, and Google Scholar search engines until June 2023. The search strategy was based on the following combinations of free text keywords and Medical Subject Heading (MeSH) terms: ”reticulate pigmentary disorders”, “dyschromatosis symmetrica hereditaria”, “dyschromatosis universalis hereditaria”, “reticulate acropigmentation of Kitamura”, “Dowling-Degos disease”, “dyskeratosis congenita”, “Naegeli-Franceschetti-Jadassohn syndrome”, “dermatopathia pigmentosa reticularis”, and “X-linked reticulate pigmentary disorder”. The Boolean operators used were “AND” and “OR ”. There were no restrictions on language. Suggested publications shown on search engines, including cited articles and citing articles, were also evaluated for appropriateness based on the titles and abstracts. Publications that were duplicates were excluded. At least two reviewers (out of M-H.L., P-C.C., I-C.L., S-F.Y., and S.Y.) screened each title and abstract to select appropriate studies for full-text review.

## 3. Dyschromatosis Symmetrica Hereditaria

### 3.1. Clinical and Histological Feature

Dyschromatosis symmetrica hereditaria (DSH, Online Mendelian Inheritance in Man (OMIM)#127400), which is also known as reticulate acropigmentation of Dohi or symmetric dyschromatosis of the extremities, is a rare pigmentary genodermatosis of autosomal dominant inheritance with nearly complete penetrance. DSH was first reported by Toyama in 1929 [3]. DSH is characterized by a mixture of hyperpigmented and hypopigmented macules on the dorsal aspects of the distal extremities [4,5,6,7,8,9]. In addition, DSH is characterized by pigmented freckle-like lesions on the face. The onset of DSH occurs most frequently in infancy or early childhood, and it may extend slowly over time, affecting the face, the sides of the neck, and the supraclavicular region [10]. Skin lesions often stop spreading before adolescence and last for life. DSH has been reported mainly in Japan, China, and Taiwan [9,11,12]. Skin biopsies for DSH demonstrate a reduced density of melanocytes in the hypopigmented macules, and the melanocytes show degenerative vacuolation, indicative of apoptosis [10]. In contrast, an increase in the melanin content of the basal layer of the epidermis is observed in the hyperpigmented macules [13]. By using split-dopa skin preparation for DSH, we demonstrated that there was a reduced density of melanocytes in the hypopigmented macules [4]. Electron-microscopic study revealed that melanocytes in the hypopigmented lesions exhibit degenerative mitochondria and cytoplasmic vacuolization [4,14].

### 3.2. Genetic Architecture

DSH is generally inherited as an autosomal dominant condition with high penetrance. Although many cases reported are familial so far, still some sporadic cases have been reported [8]. A heterozygous mutation of the *ADAR1* gene was identified to be involved in DSH [15,16]. *ADAR1* maps to chromosome 1q21.1-21.2 [17,18,19]. *ADAR1* is an RNA-editing enzyme that catalyzes the deamination of adenosine to inosine in double-strained RNA substrates during post-transcription processing [20]. Patients with DSH usually have no complications. However, recent studies showed that patients with a frameshift mutation (p.Glu673ValfsX652) in *ADAR1* presented with congenital heart disease [21] and hemangioma disease [22]. In addition, some reports showed that a mutation of p.G1007R in the *ADAR1* gene caused some neurological symptoms, such as dystonia, mental deterioration [8], and brain calcification [14]. However, in other cases with the mutation of p.G1007R in the *ADAR1* gene, no neurological abnormality was found [23]. The mechanism by which the mutations in *ADAR1* induce DSH remains to be elucidated. 

## 4. Dyschromatosis Universalis Hereditaria

### 4.1. Clinical and Histological Feature

Dyschromatosis universalis hereditaria (DUH) is a rare pigmentary genodermatosis. DUH was first report reported by Ichikawa and Hiraga in 1933 [13]. Dyschromatosis universalis hereditaria (DUH) is characterized by mottled hyperpigmented and hypopigmented macules of irregular size and shape distributed randomly all over the body [24] (Figure 1). DUH has been reported in East Asia, including China, Japan, Taiwan, and India [15,24,25,26]. DUH typically manifests during infancy or early childhood, emerges after puberty, and endures through one’s lifetime without remarkable alterations in color and distribution [27]. Histological features of a hyperpigmented skin lesion showed a pigmented basal layer of the epidermis, melanin incontinence in the papillary dermis, and melanophages and lymphocytes in the upper dermis [26]. These results indicate that DUH is not a disorder of melanocyte number. In some patients, abnormalities of hair, nails [25], and oral mucosa [28] can be observed. 

### 4.2. Genetic Architecture

DUH can be divided into three subtypes, i.e., DUH1 (OMIM#127500), DUH2 (OMIM#612715), and DUH3 (OMIM#615402) [29]. This classification is based on the different linkage regions located in the chromosomes 6q24.2-q25.2, 12q21-q23, and 2q35, respectively. DUH1 and DUH3 are inherited in autosomal dominant pattern; in contrast, DUH2 belongs to the autosomal recessive manner [29]. 

In 2013, three heterozygous missense mutations of the *SASH1* gene were first identified in Chinese DUH1 families [30]. At the same time, another group found three heterozygous missense mutations of the *ABCB6* gene in a Chinese family and sporadic patients with DUH3 [25]. It has been reported that there is a distinct clinical phenotype difference between *SASH1* mutations and *ABCB6* mutations in patients with DUH [31,32]. Very recently, genetic screening revealed that a heterozygous missense mutation; p.Q518P in the *SASH1* gene [31], with the heterozygous *SASH1* c.1547G > A mutation and *SASH1* c.1547G > T mutation [33]; and a missense mutation, c.1529G > A in the *SASH1* gene [29], were identified. DUH2 was mapped to a locus on chromosome 12q21-q23, but the specific gene has not been identified yet [34].

## 5. Reticulate Acropigmentation of Kitamura

### 5.1. Clinical and Histological Feature

Reticulate acropigmentation of Kitamura (RAK, OMIM#179850) is a genodermatosis of autosomal dominant inheritance with high penetrance [16]. RAK was first reported by K. Kitamura, S. Akamatsu, and K. Hirokawa [35]. RAK manifests as a well-defined reticulate, with slightly depressed, brownish macules affecting the dorsal aspect of hands and feet in the first or second decade of life. The macules gradually darken and enlarge with increasing age until middle age [36,37]. The typical mature-skin manifestation of RAK reveals an angular reticulate, with freckle-like hyperpigmented macules distributed on the dorsal aspect of the extremities. The macules are usually slightly depressed. About 50% of Japanese RAK patients demonstrate punctate pits on the palms and soles. Dermoscopic observation of the hyperpigmented lesions revealed a dark brown background with overlying black dots [38]. Histopathologically, RAK lesion is characterized by epidermal atrophy, with elongation and thinning of rete ridge with a slightly increased number of melanocytes and hypermelanosis [16,37].

### 5.2. Genetic Architecture

RAK is characterized as an autosomal dominant disease with high penetrance, which is caused by mutations in the *ADAM10* gene. *ADAM10* maps to chromosome 15q21.3. *ADAM10* encodes a zinc metalloprotease, which is a member of a disintegrin and metalloprotease (ADAM) family. The types of mutations in the *ADAM10* gene include the nonsense, missense, and splice-site mutations [16]. The ADAM10 protein is involved in a variety of biological processes, including regulation of the distribution patterns and transport processes of melanosomes in keratinocytes [39]. Additionally, ADAM10 is known to be involved in the ectodomain shedding of Notch proteins as substrates in the skin [35]. In a hairless mice model, an increase in skin pigmentation was noticed during aging, which was associated with the *ADAM10* mutation. These results provide an explanation for the inhibitory effect of the ADAM10 protein on melanocyte expansion [40,41].

## 6. Dowling-Degos Disease

### 6.1. Clinical and Histological Feature

Dowling-Degos disease (DDD) is characterized by acquired reticulate, with dot-like hyperpigmentation of flexures involving the axilla, submammary folds, inguinal folds, and neck. The pigmentation is usually symmetric and progressive. DDD has been reported in different regions but is more common in Caucasians [35,42]. Follicular DDD, a distinct type of DDD, presents uniquely with characteristics such as hyperkeratotic follicular papules resembling comedones; acne-like scars with a pitted appearance around the mouth; and perianal reticulated pigmented lesions, which can be observed during adulthood [43]. Galli–Galli disease (GGD) is regarded as an acantholytic variant of Dowling-Degos [44]. It is characterized by the diagnostic features of DDD, with additional acantholysis in suprabasal [45], antler with melanin at the tips, while there is no increase in the number of melanocytes. Additionally, some reports showed that DDD may manifest in a generalized form [46]. However, the genetic basis of this kind of DDD still needs more investigation to elucidate.

### 6.2. Genetic Architecture

DDD, which is also known as a reticular pigmented anomaly of the flexural surfaces [47], is characterized as a rare autosomal dominant genodermatosis caused by mutations in the *KRT5*, *POFUT1*, *POGLUT1,* and, most recently, *PSENEN* genes [48]. DDD can be classified into four subtypes. DDD1 (OMIM#179850) is caused by mutations in the *KRT5* gene on chromosome 12q13 [49]. DDD2 (OMIM#615327) is caused by mutations in the *POFUT1* gene on chromosome 20q11 [50]. DDD3 (OMIM#615674) is caused by mutations mapped to chromosome 17p13.3 [51]. DDD4 (OMIM#615696) is caused by mutations in the *POGLUT1* gene on chromosome 3q13 [52]. GGD can be found with mutations in the *KRT5* gene. GGD can also be due to *POGLUT1* mutations, and most of these cases are of European ancestry, although a Japanese patient with GGD due to a mutation in *POGLUT1* has also been reported [42]. Mutations in genes affecting melanosome trafficking and the Notch receptor signaling pathway, which is an essential regulator of melanocyte and keratinocyte proliferation and differentiation [53], have been implicated in the pathogenesis of DDD. A distinctive syndrome of Dowling-Degos with HS (HS-DDD) (OMIM#613736)) has been identified, which is caused by heterozygous variants in *PSENEN* on 19q13 [54]. A study observed that, in the same family, only obese members carrying the pathogenic *PSENEN* c.62-1G >C splice variant developed HS-DDD, while their lean family members carrying the same mutation manifested DDD only [55]. The research group also observed that, in another family, nonsmoking, lean patients having the *PSENEN* 84_85insT variant developed DDD without HS [55]. These findings suggest that environmental factors, such as obesity and smoking, can influence phenotype of patients with the same genotype and highlight that environmental factors play a role in clinical manifestations of genetic predispositions, as environmental factors, such as obesity, diet, and infection, have been identified as predisposing or exacerbating factors in certain skin immunologic diseases, such as psoriasis [56,57,58,59].

## 7. Dyskeratosis Congenita

### 7.1. Clinical and Histological Feature

Dyskeratosis congenita (DKC), which is also known as Zinsser-Cole-Engman syndrome, is characterized by congenital reticular hyperpigmentation, especially on the neck and chest, with leukoplakia and nail atrophy in fingernails and toenails. However, the classic triad of a triad of reticulated hyperpigmentation, dystrophic nails, and mucosal leukoplakia is not always observed in all individuals [60]. It is a hereditary disease that occurs predominantly in males, and the onset of DKC usually occurs in childhood between the ages of five and ten years [61]. DKC has been reported in many ethnic groups [42]. DKC manifests not only cutaneous features but also hematologic abnormalities, such as bone marrow failure, increasing risk of malignancies, pulmonary complications, and liver diseases [62]. The causative mutations of DKC are present in components of the telomerase complex. Some reports indicate that DKC is caused by defective telomere maintenance, which affects the proliferation of epithelial and hematopoietic cells and eventually leads to cellular senescence. Pigmentary changes in DKC are also attributed to increased melanin synthesis occurring in senescent melanocytes.

### 7.2. Genetic Architecture

DKC can be classified into three modes of inheritance: X-linked recessive, autosomal dominant, and autosomal recessive [63,64]. Currently, there are 14 types of DKC identified, while only 10 genes are responsible for the 14 types of DKC. Mutations in the same gene may transmit in an autosomal dominant or autosomal recessive pattern and therefore lead to different clinical manifestations and thus are defined as different types of DKC. X-linked recessive DKC (DKCX) (OMIM# 305000) is caused by the *DKC1* gene located on chromosome Xq28, which encodes for dyskerin [64], which is involved in ribosome biogenesis and in stabilizing the telomerase complex [65]. Mutations in *DKC1* mainly lead to amino acid substitutions. 

Autosomal dominant forms include DKCA2 (OMIM#613989), DKCA3 (OMIM#613990), DKCA4 (OMIM#615190), DKCA5 (Revesz Syndrome, OMIM#268130), and DKCA6 (OMIM#616553). DKCA2 is caused by mutations in the *TERT* gene on chromosome 5p15. The *TERT* gene encodes telomerase reverse transcriptase, which is the catalytic subunit of telomerase [66]. The mutated gene of DKCA3 is the *TINF2* gene on chromosome 14q12. The *TINF2* gene encodes TERF1-interacting nuclear factor 2, a critical subunit of the shelterin complex, which plays a crucial role in maintaining the length of telomeres [67]. DKCA4 is caused by mutations of the *RTEL1* gene on chromosome 20q13 [68,69]. *RTEL1* encodes the regulator of telomere elongation helicase 1, a DNA helicase which functions in the stability of telomeres [68]. DKCA5 (Revesz syndrome) is caused by mutations in the *TINF2* gene, which is also the mutated gene of DKCA3. Revesz syndrome is regarded as a rare and extremely severe DKC with extracutaneous manifestations of bilateral exudative retinopathy, bone marrow hypoplasia, aplastic anemia, cerebellar hypoplasia, and growth retardation, in addition to cutaneous manifestations of nail dystrophy, fine hairs, and reticulate skin pigmentation [70]. DKCA6 is caused by mutations in the *ACD* gene on chromosome 16q22. The *ACD* gene encodes the ACD shelterin complex subunit and telomerase recruitment factor, which is a core protein in shelterin complex and is involved in telomere function [71]. 

Autosomal recessive forms include DKCB1 (OMIM#224230), DKCB2 (OMIM# 613987), DKCB3 (OMIM#613988), DKCB4 (OMIM#613989), DKCB5 (OMIM#615190), DKCB6 (OMIM#616353), DKCB7 (OMIM#616553, the same OMIM code as DKCA6), and DKCB8 (OMIM#620133). The *NOP10* gene, also known as *NOLA3*, is located on chromosome 15q14 and encodes NOP10 ribonucleoprotein, which is a component of the telomerase complex. The first-identified autosomal recessive form of DKC was DKCB1 [72]. The homozygosity mutations were detected in 16 consanguineous families with 25 affected individuals. DKCB2 is caused by mutations in the *NHP2* gene on chromosome 5q35. *NHP2* is also known as *NOLA2* and encodes NHP2 ribonucleoprotein. The H/ACA ribonucleoprotein complex is composed of an RNA molecule and four proteins: dyskerin, GAR1, NOP10, and NHP2, and the complex is responsible for telomere maintenance [64,73]. DKCB3 is caused by mutations in the *WRAP53* gene on chromosome 17p13, which encodes an essential component of the telomerase holoenzyme complex required for telomere synthesis [74]. As in DKCA2, DKCB4 is caused by mutations in the *TERT* gene. Homozygous *TERT* mutations result in reduced telomerase activity and extremely short telomeres and can lead to a severe variant [75]. Similarly, DKCB5 is caused by homozygous or compound-heterozygous mutations of *RTEL1*, whose heterozygous mutation in the autosomal dominant form is DKCA4. The *PARN* gene is located on chromosome 16p13 and encodes poly(A)-specific ribonuclease. A mutation in the *PARN* gene causes poly(A)-specific ribonuclease deficiency that impacts telomere biology and leads to DKCB6 [76]. As mentioned earlier for DKCA6, protein products of the *ACD* gene participate in shelterin complex. Heterogeneous mutations of ACD genes from both parents have been reported and are referred to as DKCB7 [77]. The mutated gene responsible for DKCB8 is *DCLRE1B*, also known as *SNM1B*, on chromosome 1p13. *DCLRE1B* encodes the DNA cross-link repair 1B, which is involved in the repair of interstrand cross-links [78]. DKCB8 was reported by Kermasson et al. in three unrelated patients with homozygous or compound-heterozygous mutations in the *DCLRE1B* gene [79]. 

## 8. Naegeli–Franceschetti–Jadassohn Syndrome

### 8.1. Clinical and Histological Feature

Naegeli–Franceschetti–Jadassohn syndrome (NFJS, OMIM#161000) is characterized by reticulate hyperpigmentation on the neck, chest, abdomen, and axillae. Additionally, hypoplasia of dermatoglyphics, diffuse thickening of the palms and feet, hypohidrosis, dystrophy of the nails, teeth abnormalities such as enamel defects and heat intolerance owing to diminished or absent sweating are characteristic [80,81,82]. NFJS has been reported in different regions [81,83,84,85]. The histological features show that there are numerous [86]. However, the number and structure of the eccrine gland appear normal histologically.

### 8.2. Genetic Architecture

NFJS is an autosomal dominant ectodermal dysplasia caused by mutations in the *KRT14* gene on chromosome 17q11.2-q21, which codes for keratin 14 to form intermediate keratin filaments [87]. Mutations in the *KRT14* gene lead to fragility of the basal keratinocytes, which plays an essential role in the ontogeny of dermatoglyphics and sweat glands.

## 9. Dermatopathia Pigmentosa Reticularis

### 9.1. Clinical and Histological Feature

Dermatopathia pigmentosa reticularis (DPR, OMIM#125595) is characterized by reticulate hyperpigmentation located primarily on the trunk. In early literature, DPR consisted of a triad of reticulate hyperpigmentation, noncicatricial alopecia, and onychodystrophy [82]. Additionally, palmoplantar keratoderma with punctiform accentuation, widespread hyperkeratotic lesions, nail and ocular changes, ainhum formation, hypohidrosis, nonscarring blisters on the dorsal sides of the hands and feet, and pigmentation of the oral mucosa can be observed in patients with DPR [88,89]. Although the first case was reported in Switzerland, DPR has been reported in European and Asian countries [82,90]. The main differences between DPR and NFJS are that the former has lifelong cutaneous hyperpigmentation, presence of noncicatricial alopecia, and absence of dental anomalies [82,87]. The histological features showed pigmentary incontinence, vacuolar degeneration of the basal cell layer, and hyalinization of dermal collagen in the patients with DPR.

### 9.2. Genetic Architecture

DPR is an autosomal dominant condition [82] caused by mutations in the *KRT14* gene on chromosome 17q11.2-q21. Additionally, certain studies have presented findings indicating increased apoptosis in the basal cell layer, where KRT14 is expressed, upon ultrastructural examination of affected skin. This suggests that apoptosis plays a significant role in the development of DPR, providing evidence for its pathogenesis [24]. It is noteworthy that both NFJS and DPR are autosomal dominant ectodermal dysplasia syndromes due to mutations of KRT14 but have different clusters of symptoms and signs, implying variations between genotypes and phenotypes. Specifically, revertant mosaicism has been reported in epidermolysis bullosa simplex due to *KRT14* mutations [91,92,93]. It might be hypothesized that revertant mosaicism exists among clinical manifestations of the NFJS [94].

## 10. X-Linked Reticulate Pigmentary Disorder

### 10.1. Clinical and Histological Feature

X-linked reticulate pigmentary disorder (XLRPD, OMIM#301220) is a rare inherited disease first recognized by Partington et al. [95]. It is characterized by different clinical and histological features according to sex. In males, it is characterized by prominent reticulate hyperpigmentation and hypopigmentation and unique facial features, such as upswept frontal hairline and flared eyebrows, with systemic manifestations in various organs, including gastrointestinal inflammation, photophobia due to corneal opacification, recurrent respiratory infections, and failure to thrive [96,97]. Hypohidrosis is also a clinical feature [97]. Heterozygous females have milder disease than hemizygous males. In affected females, it is characterized by brown patchy pigmentary skin lesions along the lines of Blaschko, without known systemic manifestations [96,98]. The lines of Blaschko are attributed to the clonal proliferation of genetic mosaicism in keratinocytes that develop from postzygotic mutation during embryogenesis [99]. The distribution of pigmentation in female patients mimics stage 3 (hyperpigmented stage) incontinentia pigmenti, and both diseases are X-linked [100]. Other differential diagnoses of the pigmentary skin lesions along the lines of Blaschko include progressive cribriform and zosteriform hyperpigmentation (PCZH) and linear and whorled nevoid hypermelanosis (LWNH) [97]. Although the first reported case was in Canada, XLRPD has been identified in patients with different ancestry, including Maltese [101], Lebanonian [102], Korean [98], and Chinese [97]. Histopathological analysis shows mild hyperkeratosis, acanthosis, hyperpigmentation of the basal layer, and melanin incontinence in the upper dermis [96]. Electron-microscopic study shows a high number of melanosomes and some degenerating keratinocytes [102].

### 10.2. Genetic Architecture

XLRPD is inherited as an X-linked trait [97]. The disorder is caused by a recurrent intronic mutation ((NM_016937.3:c.1375–354A > G) in *POLA1*, which encodes the catalytic subunit of DNA polymerase-α [103,104]. The intronic *POLA1* mutation (c.1375–354A > G), causing altered *POLA1* gene splicing, leads to reduced transcript and protein levels [105]. *POLA1* mutations in XLRPD are associated with reduced levels of cytosolic RNA/DNA hybrids, which results in the activation of type I interferons (IFNs) and upregulation of interferon-stimulated genes. The autoinflammatory manifestations, such as gastrointestinal inflammation and keratitis, may be due to the type I IFN activation [105]. It has been reported that patients with XLRPD have decreased numbers of natural killer (NK) cells and reduced NK cell toxicity, and the defect in NK cell function may partially account for the immunodeficiency status of patients with XLRPD [105]. As the Janus kinase (JAK)/signal transducer and activators of transcription (STAT) signaling is the downstream of type I IFN receptors, JAK inhibitors have been proposed as a treatment for XLRPD. In a report, the JAK1/3 inhibitor tofacitinib improved gastrointestinal inflammation and reduced the CRP level in a patient with XLRPD [105].

## 11. Conclusions

The mutated genes and clinical manifestations of inherited reticulate pigmentary disorders are summarized in Table 1. DSH, DUH, and RAK are primarily inherited reticulate pigmentary disorders in East Asians. The pattern and distribution of pigmentation differ among these three disorders, and these clinical manifestations are useful information for differential diagnoses in East Asians with reticulate pigmentary disorders. Identification of genetic mutations, however, is the gold standard for accurate diagnosis. DDD has been reported in different regions, although most cases are of European ancestry. The characteristic reticulate, with dot-like hyperpigmentation on flexural surfaces, provides diagnostic clues, while further genetic analysis is needed for identification of subtypes. DKC, NFJS, DPR, and XLRPD show no racial predilection. DKC is a severe disorder with multiple organ involvement due to mutations that involve dysfunction of telomerase complexes. NFJS and DPR are characteristic of their association with ectodermal dysplasia, with other clinical manifestations varying between them. XLRPD is X-linked, and male patients have reticulate hyperpigmentation and hypopigmentation, while female patients have patchy pigmentation along the lines of Blaschko. Additionally, male patients with XLRPD have multiple cutaneous and extracutaneous manifestations, while female patients only have cutaneous pigmentation.

## Figures and Tables

**Figure 1 genes-14-01300-f001:**
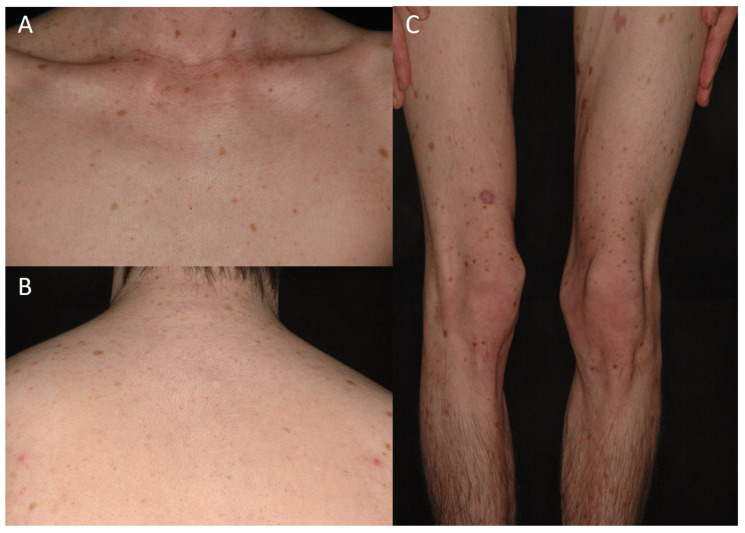
Clinical photos of a patient with dyschromatosis universalis hereditaria with generalized symmetrically distributed hypopigmented and hyperpigmented macules of varying size on (**A**) chest, (**B**) back, and (**C**) lower extremities.

**Table 1 genes-14-01300-t001:** Mutated genes, clinical manifestations, and reported regions of inherited reticulate pigmentary disorders.

Disease	Responsible Gene	Prevalent Ethnicity	Pigmentation Pattern	Other Clinical Manifestations
Dyschromatosis symmmetrica hereditarian	*ADAR1*	East Asian	mottled hypopigmented and hyperpigmented macules over the dorsal aspects of the extremities	congenital heart disease, hemangioma disease, neurological symptoms
Dyschromatosis universalis hereditaria	*SASH1* for DUH1, chromosome 12q21-q23 for DUH2, *ABCB6* for DUH3	East Asian	mottled hyperpigmented and hypopigmented macules of irregular size and shape distributed randomly all over the body	abnormalities of hair and nails
Reticulate acropigmentation of Kitamura	*ADAM10*	East Asian	angular reticulate, freckle-like hyperpigmented macules distributed on the dorsal aspect of the extremities	epidermal atrophy
Dowling-Degos disease	*KRT5* for DDD1, *POFUT1* for DDD2, chromosome 17p13.3 for DDD3, *POGLUT1* for DDD4	Caucasian	reticulate, dot-like hyperpigmentation of flexures	comedo-like follicular papules
Dyskeratosis congenita	*DKC1* for DKCX, *TERT* for DKCA2, *TINF2* for DKCA3, *RTEL1* for DKCA4, *TINF2* for DKCA5, *ACD* for DKCA6, *NOP10* for DKCB1, *NHP2* for DKCB2, *WRAP53* for DKCB3, *TERT* for DKCB4, *RTEL1* for DKCB5, *PARN* for DKCB6, *ACD* for DKCB7, *DCLRE1B* for DKCB8	no racial predilection	congenital reticular hyperpigmentation, especially on the neck and chest, with leukoplakia and nail atrophy in fingernails and toenails	hematologic abnormalities
Naegeli–Franceschetti–Jadassohn syndrome	*KRT14*	no racial predilection	reticulate hyperpigmentation on the neck, chest, abdomen, and axillae	hypoplasia of dermatoglyphics, dental anomalies, diffuse thickening of the palms and feet, hypohidrosis, nail dystrophy
Dermatopathia pigmentosa reticularis	*KRT14*	no racial predilection	reticulate hyperpigmentation located primarily on the trunk	aplasia of dermatoglyphics, noncicatricial alopecia, hypohidrosis, nail dystrophy
X-linked reticulate pigmentary disorder	*POLA1*	no racial predilection	male: reticulate hyperpigmentation and hypopigmentation; female: patchy pigmentation along the lines of Blaschko	male: upswept frontal hairline, flared eyebrows, hypohidrosis, gastrointestinal inflammation, recurrent respiratory infections, failure to thrive; female: lack of systemic manifestations

## Data Availability

Not applicable.

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
