# Peer review of "Inherited Reticulate Pigmentary Disorders"

_genes, 2023, doi:10.3390/genes14061300_

Round 1

Reviewer 1 Report

The authors describe the manuscript as a mini review, which I am  assuming is a short, narrative review. Regardless of its brevity, I think any review should include a search strategy.

Given that the manuscript describes pigmentary disorders, it would be appreciated if the text is complemented by clinical photos.

Please italicize gene names throughout the manuscript.

You have attempted to make genotype phenotype associations, it may be a good idea to tabulate your findings.

A figure mapping variants across protein domains may be interesting for the readers.

The histological feature describe in line 121-122 is sometimes referred to as “antler-like”.

In the genetic architecture section of dowling-degos, it may be interesting to mention that the condition exists as a collision disease with Hidradenitis Suppurativa (PMID: 35401657)

I would restructure the conclusion as a summary of the report (avoid presenting new or discussion of findings)

Author Response

Response to Reviewer 1

Comments and Suggestions for Authors

The authors describe the manuscript as a mini review, which I am  assuming is a short, narrative review. Regardless of its brevity, I think any review should include a search strategy.
Response: Thank you for your comments. We have added a search strategy section.

Given that the manuscript describes pigmentary disorders, it would be appreciated if the text is complemented by clinical photos.
Response: Thank you for your suggestion. We have added clinical photos of a patient with dyschromatosis universalis hereditaria

Please italicize gene names throughout the manuscript.
Response: Thank you for your comment. We have italicized gene names throughout the manuscript.

You have attempted to make genotype phenotype associations, it may be a good idea to tabulate your findings.
Response: Thank you for your suggestion. Instead of tabulating genotype-phenotype associations, we added a description about possible explanations of the associations. The description is listed below:
“It is noteworthy that both NFJS and DPR are autosomal dominant ectodermal dysplasia syndromes due to mutations of KRT14 but have different clusters of symptoms and signs, implying variations between genotypes and phenotypes. Specifically, revertant mosaicism has been reported in epidermolysis bullosa simplex due to KRT14 muta-tions[91-93]. It might be hypothesized that revertant mosaicism exists among clinical manifestations of the NFJS-DPR spectrum[94].”

A figure mapping variants across protein domains may be interesting for the readers.
Response: Thank you for your suggestion. Because the review article focuses on clinical manifestations and their genetic mutations rather than protein structure, we didn’t show such a figure in the article. Instead, we listed the mutated genes in the table.

The histological feature describe in line 121-122 is sometimes referred to as “antler-like”.
Response: Thank you for your suggestion. We have rephrased the sentence and added the “antler-like“ term in the sentence.

In the genetic architecture section of dowling-degos, it may be interesting to mention that the condition exists as a collision disease with Hidradenitis Suppurativa (PMID: 35401657)
Response: Thank you for your suggestion. We have cited the reference and discussed the association.

I would restructure the conclusion as a summary of the report (avoid presenting new or discussion of findings)
Response: Thank you for your suggestion. We have rephrased the conclusions as a summary of the report.

Reviewer 2 Report

Line 67: in another cases: in other cases

Line 76:  reported East Asia,...: reported in East Asia,...

Line 82:  we can found: we can find

Line 119: It is characterized by diagnostic features DDD with additional histopathologic finding of acantholysis...: An additional histopathologic finding is acantholysis...

Line 135: Although Japanese patient...: Although a Japanese patient...[??]

Line 162: mutations telomerase-associated proteins;: mutations in telomerase-associated proteins

Line 176: ectodermal dysplasias: ectodermal dysplasia

Line 187: are all can be observed: can be obsered

Line 188: reported in Swiss: reported in Switzerland

Line 194: ...of patient skin biopsy specimens: of affected skin

Line 205: for differential diagnoses: for differential diagnosis 

Line 207: most of cases: most cases

Line 208: is diagnostic clues: is a diagnostic clue

The quality of English is rather low. See also comments to the authors.

Author Response

Response to Reviewer 2

Comments and Suggestions for Authors

Line 67: in another cases: in other cases
Response: Thank you for your correction. We have corrected this typo.

Line 76:  reported East Asia,...: reported in East Asia,..
Response: Thank you for your correction. We have corrected it.

Line 82:  we can found: we can find
Response: Thank you for your comments. We have rephrased the sentence.

Line 119: It is characterized by diagnostic features DDD with additional histopathologic finding of acantholysis...: An additional histopathologic finding is acantholysis...
Response: Thank you for your comments. We have rephrased the sentence about Galli-Galli disease.

Line 135: Although Japanese patient...: Although a Japanese patient...[??]
Response: Thank you for your correction. We have corrected the sentence.

Line 162: mutations telomerase-associated proteins;: mutations in telomerase-associated proteins
Response: Thank you for your correction. We have corrected it.

Line 176: ectodermal dysplasias: ectodermal dysplasia
Response: Thank you for pointing out the typo. We have corrected it.

Line 187: are all can be observed: can be observed
Response: Thank you for pointing out the incorrect grammar. We have rephrased it.

Line 188: reported in Swiss: reported in Switzerland
Response: Thank you for pointing out misuse of the word. We have corrected it.

Line 194: ...of patient skin biopsy specimens: of affected skin
Response: Thank you for your suggestions: We have rephrased the sentence.

Line 205: for differential diagnoses: for differential diagnosis
Response: Thank you for pointing out the typo. We have corrected it. 

Line 207: most of cases: most cases
Response: Thank you for your correction. We have corrected the sentence.

Line 208: is diagnostic clues: is a diagnostic clue
Response: Thank you for pointing out the grammar error. We have corrected the sentence.

Comments on the Quality of English Language

The quality of English is rather low. See also comments to the authors.
Response: Thank you for your comments. We have sent the manuscript to a native English speaker to check the English language.

Round 2

Reviewer 1 Report

The authors have made extensive revisions to the text.

English language is fine and only requires minor attention.